

# Association of trait and specific hopes: cross sectional study on students and workers of health professions in Split, Croatia

Mario Maličeki, Domagoj Marković and Matko Marušić

School of Medicine, Department of Research in Biomedicine and Health, University of Split, Split, Croatia

## ABSTRACT

**Introduction.** Hope (hoping) is most commonly assessed as a dispositional trait and associated with quality of life, self-care agency and non-attempts of suicide. However, little research has been conducted on hoping for specific events.

**Materials and Methods.** We distributed a survey consisting of Integrative Hope Scale (IHS) and visual analogue scales on which respondents could declare their levels (intensity) of hope for specific events, to all first year health students enrolled at the University Department of Health Studies, Split, Croatia in 2011/2012, as well as to working health professionals attending a nursing conference in April 2012.

**Results.** A total of 161 (89.4%) students and 88 (89.8%) working health professionals returned the completed questionnaires. We found high trait hope scores of students and working health professionals (Md = 111, 95% CI [109–113] vs. Md = 115, 95% CI [112–119]; $U = 5,353$, $P = 0.065$), and weak to moderate correlations of trait and specific hopes ($r = 0.18$–$0.48$, Spearman's rank correlation coefficient). Students and workers reported 31 different things they hoped for most in life, of which the most prevalent were being healthy and happy. There was very little agreement between participants' reported influence of the four factors compromising the trait hope (self-confidence, ambition, optimism, and social support) on their specific hopes.

**Conclusions.** Our findings, while strengthening the validity of hope as a trait, indicate that specific hopes of individuals are moderated by factors not captured by the IHS trait scale. Further research should explore specific hoping in detail, as well as the effectiveness of interventions aimed at increasing specific or generalized hoping.

Corresponding author
Mario Maličeki,
mario.malicki@mefst.hr

## INTRODUCTION

Hope (hoping) is regarded as the earliest and the most indispensable virtue inherent in the state of being alive (*Erikson, 1964*). It is the central tenet of religions, especially Christianity (*Benedict XVI, 2006*; *Titus 1:2, 2011*), and an indispensable companion of illness and healing. It accompanies researchers during their scientific discoveries and individuals during their tribulations. Hope has been a popular topic in literature and arts,

ever since its entrapment in Pandora's box (*White, 1914*); and recently, it has become the topic of growing research in the fields of positive psychology, philosophy, nursing and medicine (*Cutcliffe & Herth, 2002*; *Kylma & Vehvilainen-Julkunen, 1997*; *Schrank, Stanghellini & Slade, 2008*; *Smith, 2012*; *Snyder et al., 1996*). Hope has been positively correlated with quality of life (*Evangelista et al., 2003*), self-care agency (*Alberto & Joyner, 2008*), caregiver burden (*Zink Jadaa, 2008*), and non-attempts of suicide (*Meadows et al., 2005*). However, its measurement and conceptualization is still a topic of great debate (*Boyd, 2015*; *Bright et al., 2011*; *Kylma & Vehvilainen-Julkunen, 1997*; *Lopez & Snyder, 2003*; *Schrank, Stanghellini & Slade, 2008*). In short, although hope is widely perceived as something that can be higher for one object or event than for another and that can fluctuate in its intensity, thresholds and norms for specific hopes in populations, or in patients affected or recovering from serious illnesses, have not been explored or measured. Researchers have instead focused on qualitatively identifying factors that generate or quell hope (*Soundy et al., 2014*), or have focused on quantitatively measuring hope, either as an universal (trait) that applies across situations and times; or more specifically as state hope, a person's current hoping disposition (*Lopez & Snyder, 2003*; *Snyder et al., 1996*). More than 32 instruments for the measurement of hope have been developed, and recently researchers have combined the properties of the most commonly used instruments (Miller Hope Scale, Herth Hope Index, Snyder Hope Scale) into an Integrative Hope Scale (IHS) (*Schrank et al., 2011*). It was the goal of our research to determine the association of the universal (trait) hope, measured by the IHS, with hoping for specific events, measured by declaring the intensity (level) of hope on visual analogue scales. Additionally, to further determine the relationship between the universal and specific hoping, we explored the congruency between the strongest scoring factor of the IHS trait scale (confidence, positive future orientation, lack of perspective, social relations) and the participants' perception regarding which factor influenced their specific hoping the most.

## MATERIALS AND METHODS

### Questionnaire

The English version of the IHS had been translated into Croatian by the authors and then back translated by an independent language expert to confirm its validity. Four items were reformulated in the process. Alongside demographical questions on age and sex, we also asked the participants to declare the level (intensity) of their specific hopes on the visual analog scale (VAS), graded from 0 to 100 (with every 10 intervals marked), for two different events: finishing their studies in time and being healthy at the age of 60. We then asked the respondents to name (using an open ended question) what they hope for most in life, and to designate their level of hope for that stated goal. Following each of the VAS questions we also asked the respondents to list the four factors: self-confidence, ambition, optimism, and social support; from most to least contributing to their previously stated level of hope (Appendix S1). We chose these four factors as they compromised the IHS subscales (factors): 'trust and confidence,' 'positive future orientation,' 'lack of perspective,' 'social relations and personal value'

(*Schrank et al., 2011*). As the stated goals were positive, we found that 'optimism' as a term best captures the inverse of the 'lack of perspective' subscale.

## Sampling and procedures

We used two-stage convenience sampling of two different age groups of health professionals. First, in order to assess if the level of hope declared on the VAS or IHS could be influenced by the order by which examinees filled out these questionnaires, we randomized all first year students of health studies at the University of Split (who enrolled in their first year of studies in 2011/2012) into two groups: the 1st group was given the IHS questionnaire followed by the VAS, while the 2nd group was first given the VAS followed by the IHS. A simple random number generator was used for random allocation to the groups. As we found no evidence that the order of presenting questionnaires influenced either IHS or VAS scores (Table S1), in further analysis we treated both groups as one. Additionally as the student population was age-homogeneous, in order to check for the possible influence of age on IHS or VAS scores, we administered the questionnaire to the working health professionals who attended the Education for lecturers of nursing courses in April 2012, Split. All of the working health professionals were given a questionnaire in which the IHS questionnaire was printed first. Cronbach's alpha of the IHS for both groups combined was 0.869 (95% CI [0.843–0.892]) showing good internal consistency.

## Statistical analysis

Frequencies and percentages were used for the description of categorical variables, and median (Md) and interquartile range (IQR) for non-normal distributions. The Mann–Whitney $U$ test was used to assess the difference in medians between the groups, while the chi-square test was used to compare frequency distributions of categorical variables. Correlations between the IHS total and subscale scores with VAS scores were assessed by Spearman's rank correlation coefficient. Concordance of the ranking order with which the participants graded factors which influenced their hope levels were determined using Kendall's coefficient. The level of significance for all statistical tests was 0.05. Data was analyzed with SPSS statistical package 19.0 (SPSS; Chicago, Illinois, USA).

## Ethical approval

The study was approved by the ethical review board of University of Split, School of Medicine, Croatia (no. 003-08/11-03/0005).

## RESULTS

### Demographic data

A total of 161 (89.4%) students of first year health studies (132 women, 26 men, missing data for 3 respondents) participated in the study, as well as 88 (89.8%) working health professionals attending a nursing conference (86 women, 2 men). The students were 18–47 years old, with a median age of 19 (IQR = 19–21), and the workers were 22–70 years old, with a median age of 48 (IQR = 38–52).
**Table 1 Integrative hope subscale scores and levels of hope designated on visual-analog scales (VAS) for students ($n = 161$) and workers ($n = 88$) of health professions.**

| Variable | Students median (IQR) | Workers median (IQR) | $P^a$ |
|---|---|---|---|
| **Integrative hope total score** | 111.0 (105–118) | 115.0 (106–121) | 0.065 |
| **Integrative hope subscale** | | | |
| Trust and confidence | 32.5 (31–36) | 36.0 (32–38) | <0.001 |
| Lack of perspective | 27.0 (24–30) | 26.0 (24–30) | 0.653 |
| Positive future orientation | 27.0 (25–29) | 27.0 (25–29) | 0.873 |
| Social relations and personal value | 24.0 (22–26) | 26.5 (23–28) | <0.001 |
| **Levels of hope on VAS for** | | | |
| Finishing studies in time | 90 (80–100) | / | / |
| Being healthy at the age of 60 | 70 (60–80) | 85 (70–92) | <0.001 |
| The most hoped-for thing in life | 90 (76–100) | 95 (80–100) | 0.041 |

Notes.
<sup>a</sup>Mann–Whitney $U$ test.

**Table 2 Correlation of trait hope, specific hopes and age of students ($n = 161$) and workers ($n = 88$) of health professions.**

| Correlation ($\rho^a$, 95% CI) | Hope for finishing studies in time | Hope for being healthy at the age of 60 | Most hoped-for thing in life | Age |
|---|---|---|---|---|
| Students' trait hope | 0.275 (0.124–0.413) | 0.182 (0.027–0.328) | 0.318 (0.169–0.452) | 0.0261 (−0.130–0.181) |
| Workers' trait hope | / | 0.421 (0.210–0.595) | 0.486 (0.278–0.650) | −0.0140 (−0.340–0.0732) |

Notes.
<sup>a</sup>Spearman's rank correlation coefficient.

## Comparison of students and working health professionals

There was no significant difference between the two groups in their IHS total score (Md = 111, 95% CI [109–113] vs. Md = 115, 95% CI [112–119]; $U = 5,353$, $P = 0.065$). However, workers had higher scores on the IHS' 'trust and confidence' and 'social relations and personal value' subscale scores, as well as higher hopes (designated on VAS) of being healthy at the age of 60 and for the things they most hoped for in life (Table 1).

Sex differences were observed for the student population, with males reporting higher hopes for being healthy at the age of 60 ($U = 1153.5$, $P = 0.009$).

For both groups, universal (trait) hope, measured by the IHS, showed a significant strength of correlation ($r = 0.18$–$0.48$) with specific hopes, measured by the VAS (Table 2).

When answering an open ended question on what their most hoped-for thing in life was, students and workers listed 1–5 answers, with no differences between the groups on the number of answers they listed (Md = 1, 95% CI [1–2] vs. Md = 2, 95% CI [1–2], $U = 5,373$, $P = 0.169$). Cumulatively, 31 most hoped-for concepts emerged, with health and happiness being the most prevalent in both groups. However, the frequency distribution of individual concepts showed several significant differences, with students hoping more for health, work and family, while workers hoped more for life contentment (Table 3).

**Table 3** Concepts that students ($n = 157$) and workers ($n = 78$) listed as their answers to the question: "What do you most hope for in life?"

| Answer | No (%) of | | |
| --- | --- | --- | --- |
| | Students | Workers | $P^{a}$ |
| Health | 73 (46.50) | 55 (70.51) | <0.001 |
| Happiness | 39 (24.84) | 17 (21.79) | 0.7237 |
| Work/carrier | 38 (24.20) | 8 (10.26) | 0.0181 |
| Family | 29 (18.47) | 6 (7.69) | 0.0465 |
| Love | 10 (6.37) | 11 (14.10) | 0.0865 |
| To finish studies | 9 (5.73) | / | 0.0726 |
| To achieve my goals | 7 (4.46) | 1 (1.28) | 0.3775 |
| Money | 6 (3.82) | 3 (5.13) | 0.7251 |
| Children | 4 (2.55) | 2 (2.56) | 0.6660 |
| Marriage | 4 (2.55) | / | 0.3754 |
| Living | 4 (2.55) | 3 (3.85) | 0.8856 |
| Winning a lottery | 2 (1.27) | / | 0.8049 |
| Peace | 2 (1.27) | 4 (5.13) | 0.1852 |
| To be content | 1 (0.64) | 5 (6.41) | 0.0276 |
| Advanced age/longevity | 1 (0.64) | 4 (5.13) | 0.0773 |
| Spiritual fulfillness | 1 (0.64) | 4 (5.13) | 0.0773 |
| Children's happiness | / | 2 (2.56) | 0.2073 |
| Mingling | / | 2 (2.56) | 0.2073 |
| Other[b] | 1 (0.64) | 1 (1.28) | 0.8049 |

Notes.

[a] Chi-square test.

[b] Includes concepts: to remain the same, to have no worries, everything, helping family members, good grades, good relationship with colleagues, fun, food, knowledge, social security, grandchildren, for no tragedies in life.

## Influence of hope trait factors on specific hoping

After designating levels of hope on the VAS scales participants declared how much the four factors (self-confidence, ambition, optimism, and social support) contributed to the levels of hope they designated. The same order of the factors was listed by 23 (14.3 %) students, and 0 (0%) workers. The order of the factors between different participants showed very little agreement, even when participants with highest or lower trait hope scores were analyzed separately (Kendall's W from 0.024 to 0.117; Table S2).

Of the four factors, optimism was most commonly chosen by the participants of both groups as the factor which contributed most to the hope of being healthy at the age of 60, as well as for their most hoped-for thing in life ($\chi^2 = 2.632$, $P = 0.004$ and $\chi^2 = 6.438$, $P = 0.09$, respectively). No single factor was chosen by the students as that which contributes most to their hope of finishing studies in time, but rather all four factors (self-confidence, ambition, optimism, and social support) were represented in equal measure ($\chi^2 = 6.903$, $P = 0.075$, Table S3).

In order to see if the factor which individuals chose as the most influential to their specific hopes was also the one with the highest score on the IHS (sub)scale, we ranked the IHS subscales scores of each individual from highest to the lowest. This resulted in ambition (positive future orientation) being expressed as the strongest factor of the four for both groups of participants (Tables S2 and S3).

## DISCUSSION

Our study showed that there were no differences between total scores of universal (trait) hope, measured by an Integrative Hope Scale, between training and working health professionals; and that the trait hope was weakly to moderately correlated with the intensity (level) of hope for specific events, declared on visual analogue scales. These findings strengthen the validity of hope as a human trait, and imply its stability through time, as also indicated by *Schrank et al. (2011)* on the general population of Austria. The IHS scores in our sample were however higher than those found in Austria suggesting either cultural or quality of life differences, or even the specifics of the caring profession which our sample was based on. *Averill, Catlin & Chon (1990)* have shown that religion, specifically Judeo–Christian influences on the Western nations, compared to Confucianism influences on the Eastern nations, left a profound influence on both the conceptual grasping and importance of hope. Although there is a higher percentage of declared Catholic population (86.28%) in Croatia than in Austria (73.66%) (*Croatian Bureau of Statistics, 2013*; *Statistics Austria, 2001*), neither the (*Schrank et al., 2011*) study nor our study, checked for religious orientation, requiring that these differences be explored in further studies.

Higher levels of the subscales '*trust and confidence*' and '*social relations and personal value*' of working health professionals in our study compared to those of the student population, most likely result from age specific developmental characteristics and family status. Similarly, the differences observed in the most hoped-for things in life for these two populations could originate from the higher number of individuals within the working population who have already achieved their hopes and goals for work and family, and are therefore more oriented toward life contentment and spiritual fulfillment. Workers' higher levels of hope for being healthy at the age of sixty could result from the facts that our sample consisted only of an active working population and that the workers were also closer to the 60 year-mark, meaning that they could, based on their age and health so far, better evaluate their future health. Our findings of male students having higher hopes for being healthy at the age of 60 could originate from observed gender differences in the perception of health (*Suris, Parera & Puig, 1997*); however, as our sample included only a small number of male students ($n = 26$), this difference needs to be confirmed in further studies.

Our study also adds further support for hope being an emotion that can be expressed and recollected (*Smith, 2012*), as the most hoped for things in life our participants listed are almost identical to those in the Averill's study of analysis of hope (*Averill, Catlin & Chon, 1990*), in which, wanting to "eliminate" abstract hopes, researchers asked participants to name events in the previous year when they specifically hoped for something (after having been asked to explain and provide examples of differences between wanting or desiring something, and hoping for something).

We acknowledge that our sample was not random; however, it was not the goal of this study to determine hope norms for the Croatian population, nor have such studies on hope been conducted anywhere in the world. Likewise, the most hoped-for things

in life listed by the students and workers of health professions should not be taken as representative, outside perhaps health professions, as hopes and life goals depend on a multitude of factors, including those intrinsic, generational, social and cultural (*Grouzet et al., 2005*; *Twenge, Campbell & Freeman, 2012*).

The positive association we found between a person's trait hope and their levels of hope for different specific events, coupled with the weak to moderate strength of those correlations ($r = 0.18$–$0.48$) and the differences in which individuals ranked four factors compromising trait hope (self-confidence, ambition, optimism, and social support) according to how much they contributed to their levels (intensity) of hope for specific events (Kendall's W from 0.002 to 0.15), indicates that specific hopes in individuals are most likely mediated by factors that do not compromise the IHS trait instrument. As determination and increased goal oriented actions are invoked by the changes in the intensity of hope(ing) for that goal (*Averill, Catlin & Chon, 1990*), and multiple factors have been found to influence hoping of patients recovering from stroke or spinal cord injuries (*Soundy et al., 2014*) further research should focus on determining the most influential factors for specific hopes, especially ones associated with better health outcomes (*Van Allen et al., 2015*). Additionally, effectiveness of interventions aimed at increasing specific or generalized hoping should be assessed.

## ACKNOWLEDGEMENTS

We would like to thank all the students of the 2011/12 generation of health studies in Split, as well as the attendees of the 2012 Education for lecturers of nursing courses in Split for their participation in the study. We would also like to thank Linda Ivas, Ana Utrobičić, Adrijana Banožić and Ana Jerončić for their input with questionnaire design and implementation, and to Ana Marušić for her help in reviewing our manuscript.

### Funding

The authors received no funding for this work.

### Competing Interests

The authors declare there are no competing interests.

### Author Contributions

- Mario Maličnki conceived and designed the experiments, performed the experiments, analyzed the data, contributed reagents/materials/analysis tools, wrote the paper, prepared figures and/or tables, reviewed drafts of the paper.
- Domagoj Marković performed the experiments, contributed reagents/materials/analysis tools, wrote the paper, reviewed drafts of the paper.
- Matko Marušić conceived and designed the experiments, performed the experiments, contributed reagents/materials/analysis tools, wrote the paper, prepared figures and/or tables, reviewed drafts of the paper.

## Human Ethics

The following information was supplied relating to ethical approvals (i.e., approving body and any reference numbers):

Ethical review board of University of Split, School of Medicine (no. 003-08/11-03/0005).

## Data Availability

The raw data has been supplied as Data S1.

## Supplemental Information

Supplemental information for this article can be found online at http://dx.doi.org/10.7717/peerj.1604#supplemental-information.

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
