# Peer review of "Association of trait and specific hopes: cross sectional study on students and workers of health professions in Split, Croatia"

_PeerJ, doi:10.7717/peerj.1604_

## Round 0.1 · original submission · Major Revisions

I have received two expert reviews on your manuscript. Both reviewers note that the manuscript would benefit from greater theoretical grounding in the Introductory content and in the Discussion. I hope you will respond to their critiques and submit a revised version of the document.

·

Basic reporting

Authors should refer to a conceptual hope model, in describing their variables, to be able to design the hypotheses.

The hypothesis are not clear, authors are invited to add a section to describe the main goal of their paper.

Experimental design

Hypothesis. The authors should specify at the end of the introduction the hypothesis of mediation and moderation model.
Instruments. The authors should insert for each measures the factorial structure and internal consistency reliability that measures present in their study.
Procedures. Authors should insert procedure section after measures section.

Validity of the findings

Authors should review the results in line with the hypothesis and in line with data analysis description
Does the first hypothesis analyze differences in the four subscale of IHS between students and workers they should report?
What is the second one?

Discussion. Authors are encouraged to review discussion in line with theoretical model and revised theoretical framework.

Additional comments

Table and Figures. Table and figures are not in line with APA style

·

Basic reporting

- The use of hope in non-suicide attempts needs further exploration. It is incongruent with the rest of the literature. Line 56.
- Review of Jevne's work on hope (see hope-lit database) may assist in exploring the core issue of the intangabiliities of hope that could strengthen the literature review section.

Experimental design

- Identification of why this hope scale was chosen over others is required (line 72)
- line 83 notes 'aiming for positive factors' could this perception that hope is a positive, future oriented practice skew the results? Hope can also be identified in previous literature as a burden or a 'foolish counsellor' (plato) its important for the researchers not to define hope based on perception alone.

Validity of the findings

- line 158 - could hope be viewed in terms of its stability or just persistent inclusion?
- references to religiosity and hope (line 164 and beyond) need to be noted in the lit review section.

Additional comments

The findings section might be better organised through use of sub-headings. The concluding statements need further exploration to strengthen the need for additional research in the field.

---

## Round 0.2 · accepted · Accept

Thank you for addressing the items listed by reviewers. It is my pleasure to accept this work for publication.